# Tailored Visions: Enhancing Text-to-Image Generation with Personalized Prompt Rewriting

## Abstract

We propose a novel perspective of viewing large pretrained models as search engines, thereby enabling the repurposing of techniques previously used to enhance search engine performance. As an illustration, we employ a personalized query rewriting technique in the realm of text-to-image generation. Despite significant progress in the field, it is still challenge to create personalized visual representations that align closely with the desires and preferences of individual users. This process requires users to articulate their ideas in words that are both comprehensible to the models and accurately capture their vision, posing difficulties for many users. In this paper, we tackle this challenge by leveraging historical user interactions with the system to enhance user prompts. We propose a novel approach that involves rewriting user prompts based a new large-scale text-to-image dataset with over 300k prompts from 3115 users. Our rewriting model enhances the expressiveness and alignment of user prompts with their intended visual outputs. Experimental results demonstrate the superiority of our methods over baseline approaches, as evidenced in our new offline evaluation method and online tests. Our approach opens up exciting possibilities of applying more search engine techniques to build truly personalized large pretrained models.

## 1 Introduction

We are training increasingly large and powerful foundation models (Brown et al., 2020; Rombach et al., 2022; Gal et al., 2022) through self-supervised learning. These large pretrained models (LPMs) serve as efficient compressors (Delétang et al., 2023), condensing vast amounts of internet data. This compression enables the convenient extraction of the knowledge encoded within these models via natural language descriptions. Despite being in its infancy, this approach exhibits the potential to surpass traditional search engines as a superior source for knowledge and information acquisition.

In this study, we take a fresh perspective, seeing LPMs as if they are search engines. This lens uncovers a goldmine of techniques previously utilized in enhancing search engine performance, which can be repurposed to refine foundational models. For instance, strategies for improving query formulation (Antonellis et al., 2008), data source purification(Davis, 2006; Yang et al., 2023a), and ranking optimization (Brin & Page, 1998; Song et al., 2023) can be leveraged.

To bring this concept to life, we experiment with an enhanced query generation technique in this paper. This technique restructures the query using personalized information to augment the performance of text-to-image generation models(Rombach et al., 2022), a major category of foundational models used for encapsulating image knowledge.

Akin to refining queries for search engines, prompts given to LPMs must also be carefully crafted. However, the complexity of prompts, the unpredictability of model responses compared to traditional search engines present unique challenges. Significant research efforts (Jagerman et al., 2023; Zhong et al., 2023) have been made to comprehend how LPMs react to various prompts, with some studies examining the feasibility of rewriting prompts for specificity. However, without access to users' personal data and behavior, tailoring the prompt to meet the user's needs accurately remains challenging.

**User Histories**

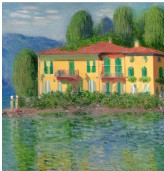

Large Italianate House with beautiful landscape on the shores of Lake Como, oil painting in the style of Claude Monet.

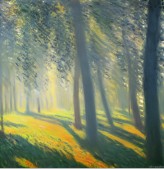

A Magical Forest in the morning by Claude Monet, oil painting in the style of Claude Monet trending in artstation.

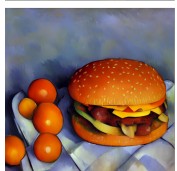

Still life of a hamburger and some fruits on a napkin. By Cezanne, detailed oil painting in the style of Paul Cezanne trending in artstation

**Original Prompt**

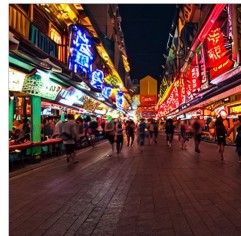

Walking street at night.

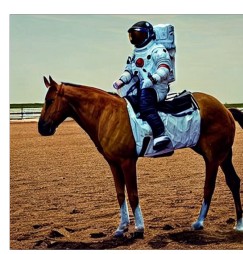

An astronaut riding a horse.

**Personalized Prompt**

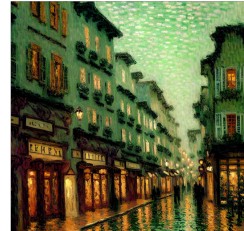

..., a mesmerizing walking street at night,..., with a touch of Italianate charm, reminiscent of Claude Monet's oil painting style.

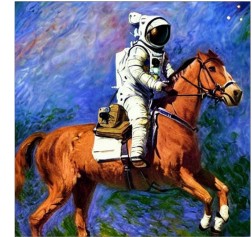

An astronaut, inspired by Claude Monet's ..., rides a horse, ..., fusion of magic and nature

Figure 1: Comparison between "Original Prompt" and "Personalized Prompt" text-to-image generation methods. **Left:** A user's histories from our PIP dataset. It intuitively implies this user favors the style of Italianate, oil painting, Claude Monet etc. **Right:** The personalized prompt rewriting results. Through the use of our personalized method, we can generate images closely align with the user's preferences, significantly different from the original prompt method and its conventional style.

Our research addresses this issue by integrating user preference information into prompt rewriting. The primary obstacle in personalized query rewriting is the absence of a dataset containing text-to-image prompts with personalized information. To overcome this, we have assembled a large dataset encompassing over 300k prompts from 3,115 users. We rewrite user prompts using their query history, although we had limited access to personal information, leaving room for further research. Another significant challenge is the evaluation of rewritten queries. To evaluate their efficacy, we've developed a new offline method that uses multiple metrics to measure how well our rewriting models can recover the original user query from the ChatGPT-shortened version.

Our paper's contributions are manifold:

1. We propose viewing large foundational models through the lens of search engines.

2. We demonstrate how this perspective can enhance human-model interaction by borrowing query rewriting ideas from the search engine community.

3. We've compiled a large Personalized Image Prompt (PIP) dataset, which will be made public upon paper acceptance to aid future research in this field.

4. We experimented with two query rewriting techniques and proposed a new query evaluation method to assess their performance.

While there is still a considerable distance to cover before we can create a perfect prompt encapsulating both the user's requirements and the model's capabilities, we believe our research provides a critical stepping stone in this ongoing exploration.

## 2 RELATED WORK

This section provides an overview of prior work on text-to-image generation, personalization for such generation, and prompt rewriting. It's important to note that our review is aimed more at offering sufficient background knowledge rather than exhaustive coverage of all related works.

## 2.1 TEXT-TO-IMAGE GENERATION

Large text-to-image generation models can generate high-fidelity image synthesis and achieve a deep level of language understanding. DALL-E (Ramesh et al., 2021) uses a VQ-VAE transformer-based method to learn a visual codebook in the first stage and then trains autoregressive transformers on sequences of text tokens followed by image tokens in the second stage. DALL-E2 (Ramesh et al., 2022) introduces latent diffusion models to generate various images by conditioning on CLIP text latents and CLIP image embeddings generated by a prior model. Imagen (Saharia et al., 2022) discovers that a larger language model with more parameters trained on text-only data improves the quality of text-to-image generation. Late developments like stable-diffusion (SD) (Rombach et al., 2022) proposes to generate images effectively in latent space significantly lowering computational costs. Furthermore, SD designs a conditional mechanism to complete class-conditional, text-to-image and layout-to-image models. Furthermore, ControlNet (Zhang & Agrawala, 2023) accomplishes certain function by conditioning on multi-modal data, e.g., edge, sketching, pose, segmentation, depth etc., which unavoidably involving additional condition-generation modalities.

Despite these models' ability to generate high-fidelity images, they often fail to meet the precise needs of the users. Text-to-image generation is more like a game of chance.

## 2.2 PERSONALIZATION FOR TEXT-TO-IMAGE GENERATION

Recently, personalization approaches based on text-to-image models have taken a set of images of a concept and generated variations of the concept. Specifically, some methods optimize a set of text embeddings. For example, Cohen et al. (2022) involves pseudo-word embeddings by a set encoder to provide personalization and Textual inversion (Gal et al., 2022) composes the concept into language sentences and performed as a personalized creation. Some methods finetune the diffusion model. For example, DreamBooth (Ruiz et al., 2023) finetunes the text-to-image diffusion model with shared parallel branches. To speed up, CustomDiffusion (Kumari et al., 2023) reduces the amount of fientuned parameters, and Tewel et al. (2023) locks the subject's cross-attention key to its superordinate category to align with visual concepts. Moreover, an additional encoder is trained to map concept images to its textual representation by Gal et al. (2023) and Shi et al. (2023).

Existing studies have three key limitations: they demand extra images and fine-tuning of text-to-image models with limited scope for new concepts; they can't learn from user interaction history and need detailed user prompts; and there's a lack of public, personalized text-to-image datasets that truly reflect user preferences.

## 2.3 PROMPT REWRITING

Recently, researchers have found that optimizing prompts can boost the performance of LLMs on several NLP tasks and even search systems. For examples, Guo et al. (2023) connect the LLM with evolutionary algorithms to generate an optimized prompt from parent prompts, without any gradient calculation. Yang et al. (2023b) propose to use LLM as an optimizer by generating new prompts based on a trajectory of previously generated prompts in each optimization step with the objective of maximizing the accuracy of the task. In the work (Zhou et al., 2022), LLMs serve as models for engineering work like inference, scoring, and resampling. In search systems, LLMs are used to generate query expansion terms by Jagerman et al. (2023), while they are used to reformulate query by Wang et al. (2023) instead.

For T2I generation, one relatively close work, SUR-adapter (Zhong et al., 2023) learns to align the representations between simple prompt and complex prompt by an adapter. Hao et al. (2022) optimize prompt through general rewriting. However, Neither of above works utilized personalized information for improving prompts.

## 3 PERSONALIZED PROMPT REWRITING

The basic idea of our personalization method is to rewrite the input prompt, considering user preferences gleaned from past user interactions. The full pipeline of our method is depicted in the left part of Figure 2. If a user $u$ input a prompt $x_t$, a retriever $\mathbf{Ret}\,(x_t, \mathcal{Q}_t)$ retrieves prompts from the

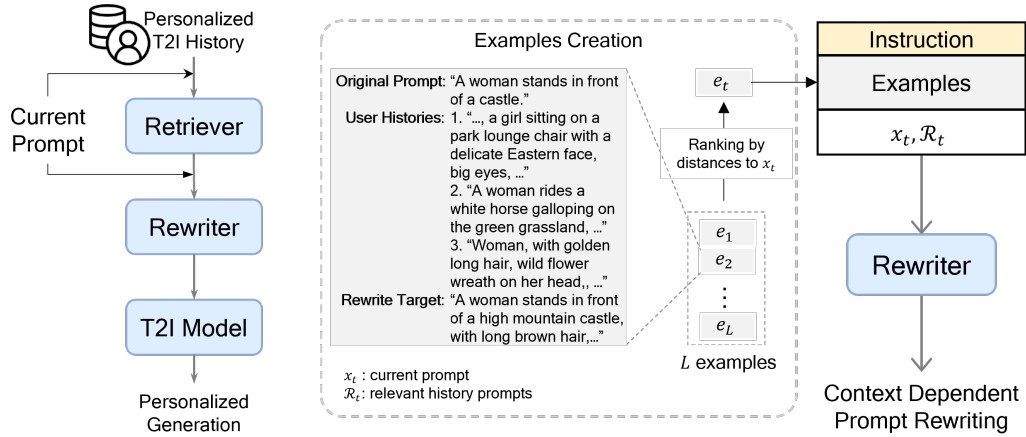

Figure 2: **Left:** Pipeline of personalized prompt rewriting, including Retriever, Rewriter and T2I model to generate personalized images from user histories. **Right:** Illustration of context-dependent prompt rewriting. We present a specific example for better understanding the procedure of context-dependent prompt rewriting.

user's historical prompt set $\mathcal{Q}_t$, using $x_t$ as a query. Based on the retrieval result $\mathcal{R}_t = \mathbf{Ret}\,(x_t, \mathcal{Q}_t)$, the rewriter $\mathbf{Rew}$ rewrite the input prompt to generate a personalized prompt $x'_t = \mathbf{Rew}\,(x_t, \mathcal{R}_t)$. Finally, the text-to-image generation model $\mathbf{G}$ produces the image $I'_t = \mathbf{G}\,(x'_t, \epsilon)$ from the rewritten prompt, where $\epsilon$ is a random noise vector or matrix[1].

## 3.1 RETRIEVAL AND RANKING

In the retrieval stage, given the input prompt $x_t$, the retriever $\mathbf{Ret}\,(x_t, \mathcal{Q}_t)$ retrieve relevant prompts from historical prompt set $\mathcal{Q}_t$, using $x_t$ as a query.

By analyzing user prompts, we have noticed that users tend to construct prompts that involve objects, their attributes, and the relationships between objects. In the works (Schuster et al., 2015), (Krishna et al., 2017), a query for image retrieval is defined to include objects, attributes of objects, and relations between objects. Inspired by this, we suspect users have the habit of using attributes and some objects, such as background, to express their preferences. To confirm this, we visualize the word cloud of the top 250 frequent words in the text prompts of all users, as shown in the right part of Figure 5. In Figure 5, we find some attributes, such as "cute", "golden", and "beautiful" appear in prompts with high frequency, as well as some objects, such as "mountain", "sea," and "sky". Intuitively, we can use the current prompt $x_t$ to locate the relevant history prompts that include the same or similar attributes or objects.

To locate relevant prompts, two retrieval methods are used: dense and sparse. In dense retrieval, we choose the prompt $x_t$ and calculate its textual embedding $\mathbf{Em}\,(x_t)$ using CLIP's text encoder, also the text encoder in Stable Diffusion (Rombach et al., 2022). We suspect the prompts with similar visual attributes and objects will be close to each other in the text embedding space. To confirm this, we visualize some retrieval results in Figure 6. The three nearest neighbors of $\mathbf{Em}\,(x_t)$ are prompts that are semantically related. For example, if the input prompt is "Hobbit homes", the three most relevant prompts would include the words "village", "city", and "house". This dense retrieval method is also referred to as embedding-based retrieval (EBR). In sparse retrieval, we use BM25 to locate relevant prompts that include the same visual attributes and objects.

In the above retrieval, we rank relevant prompts in EBR-based or BM25-based ranking, depending on the retrieval ways. In EBR-based ranking, we rank the relevant prompts based on their embedding similarity with the query $x_t$. For similarity measuring, we choose cosine similarity as it is a commonly used similarity measure in embedding learning. In BM25-based ranking, BM25 scores are used for similarity measures. Consequently, we obtain the top $k$ relevant user queries $\mathcal{R}_t = \{r_1, ..., r_k\}$.

---

[1]The image generated from the prompt $x_t$ is denoted as $I_t$, where $I_t = \mathbf{G}\,(x_t, \epsilon)$.

## 3.2 REWRITTING

The procedure of context-independent rewriting leverages pertinent queries $\mathcal{R}_t = \{r_1, ..., r_k\}$, and employs ChatGPT to encapsulate user preferences and rewrite the prompt directly. These queries $\mathcal{R}_t$ are organized based on their relevance to $x_t$. An elaborate example of this process is available in Appendix A2.

In the context-dependent scenario, we initially create a collection of demonstration examples $\mathcal{E} = \{e_1, ..., e_L\}$ using manual design. We then select a small subset of these examples to serve as demonstrations for each rewriting task. Given the issue of order sensitivity in in-context learning, as highlighted in the study (Liu et al., 2021), we arrange the demonstration examples in a descending sequence based on their proximity to the input prompt $x_t$. The in-context rewriting process we employ is illustrated in the right section of Figure 2.

## 4 PERSONALIZED IMAGE-PROMPT (PIP) DATASET

### 4.1 DATASET COLLECTION

The Personalized Image-Prompt (PIP) dataset is the first large-scale personalized generation dataset. The original data are collected from a public website that we host to provide open-domain text-to-image generation to users. To construct PIP, we select 300,237 image-prompt pairs created by 3115 users using SD v1-5 (Rombach et al., 2022) and an internal fine-tuned version of SD v1-5. We only include users who have created 18 or more images or provided at least 12 different prompts.

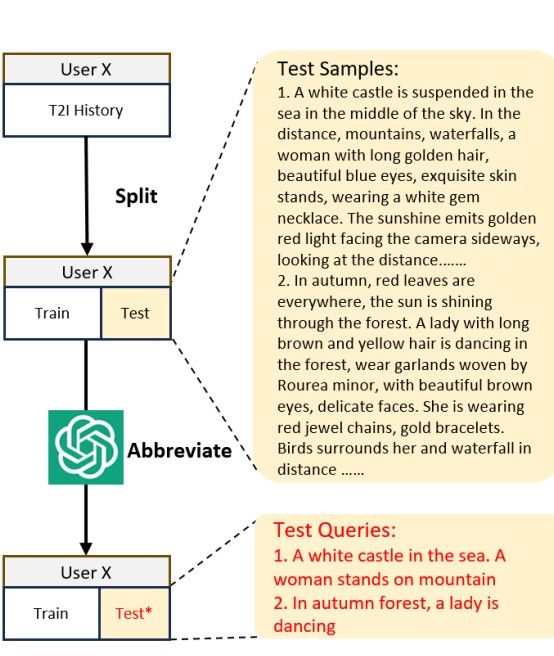

Figure 3: Data collection pipeline.

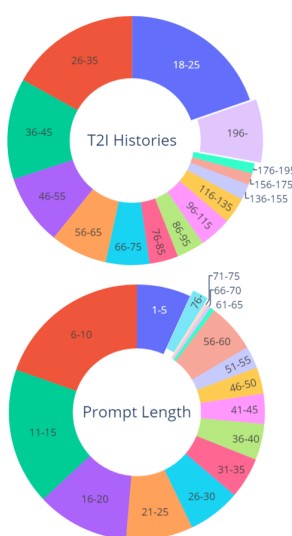

Figure 4: Dataset statistics and distribution. **Upper:** Proportion of users based on the varying number of histories they have. Note that each user has a minimum of 18 histories, as we have excluded those with fewer histories from the dataset. **Lower:** Proportion of prompts based on their varying lengths .

Figure 3 illustrates the process of creating the dataset. For each individual user, we randomly choose two prompts to serve as test prompts, with the remaining prompts allocated as training prompts (historical user query). This approach of random selection, as opposed to using the most recent generated prompts for testing, is adopted to enhance the diversity of our test data. Subsequently, we employ ChatGPT to condense the test prompts, ensuring they only include the primary object or

scene, as depicted in Figure 3. We shorten the prompts into three scales, i.e., contain only nouns, noun phases or short sentences respectively.

In the ensuing experiment, each test prompt in the test set will be considered as the input prompt $x_t$ for every user $u$, with the original prompts serving as the ground truth that mirrors the user's authentic preferences. The remaining prompts are utilized as training samples.

The PIP dataset consists of 300,237 image-prompt pairs, personally categorized by 3,115 users. These pairs are divided into 294,007 training samples and 6,230 test samples.

## 4.2 DATASET STATISTICS AND DISTRIBUTION

In this section, we showcase the data statistics that depict the quality and diversity of PIP. We specifically illustrate data distributions of the number of prompts and prompt length for each user, and delve deeper into the content of the prompt through a word cloud representation.

Each data sample contain a prompt, the generated images, UserID, Image size, and the URL, as illustrated in the left part of the Figure 5.

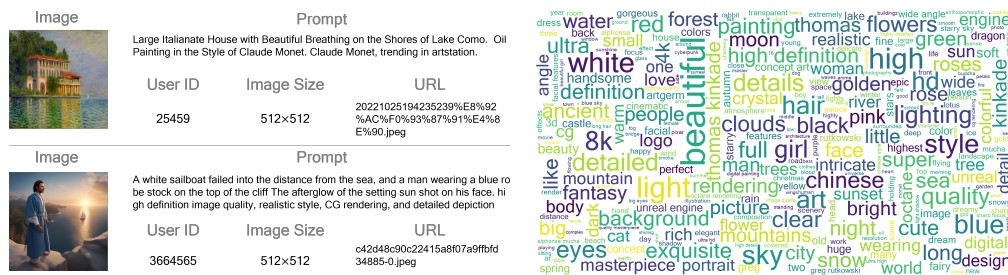

Figure 5: **Left:** Two examples from User histories, containing `Image`, `Prompt`, `User ID`, `Image size` and `URL`. **Right:** Word cloud visualization of top 250 keywords sampled from the PIP dataset.

In the PIP dataset, each user contributes at least 18 images, as depicted in the upper section of Figure 4. This results in a long-tail distribution. The prompts have an average word count of 27.53. The length of the prompts, ranging from 1 to 284 words, also follows a long-tail distribution, as seen in the lower section of Figure 4. Despite the presence of approximately 2500 prompts that surpass the 75-word limit of SD, we retain them to maintain the integrity of user preferences.

Figure 5 presents the 250 most frequently used words or phrases. The frequency of these words is determined by the highest TF-IDF value across all users. The word cloud reveals that these words describe various image attributes, such as objects, styles, quality, and colors. This variety underlines the high diversity present within the prompt content of the PIP dataset.

## 5 EXPERIMENT

We validate our personalized prompt rewriting method through both offline and online evaluation.

### 5.1 EVALUATION METHOD

In the offline assessment, we introduce a novel evaluation technique to gauge the rewriter's capability to restore the original user query from its shortened version produced by ChatGPT. An example of input is shown in Table 1. For this evaluation, we utilize all the abbreviated prompts from the PIP test set, in conjunction with the users' actual prompts and saved images.

The metrics used for this offline evaluation include:

**ROUGE-L.** Given a shortened prompt, ROUGE-L compares the rewritten prompt against the original prompt. It measures the ability of our prompt rewriter to recover the original prompt. We set $\beta = 5$ to emphasize the recall of the generated prompts.

---

**ChatGPT Prompt for Creating Test Examples**

---

Your task is to abbreviate a given text-to-image prompt. The abbreviated prompt should just include the primary object from the original prompt and basic attributes, ignoring trivial details and other descriptions. Generally speaking, an abbreviated prompt should be less than 10 words.
Please abbreviate the following prompt:
The original prompt: *Autumn leaves, detailed, concept art, low angle, high detail, warm lighting, huge scene, grass, art Greg Rutkowski, trending in artstation.*
The abbreviated prompt:

---

Table 1: An input example for test prompt abbreviation.

**Distance to Historical Prompts (DHP).** DHP measures how close the shortened prompt or the rewritten prompt is to the previous prompts in Euclidean embedding space. GTR-T5-large (Ni et al., 2021) is used for extracting textual embeddings.

**Preference Matching Score (PMS).** PMS calculates the CLIPScore (Hessel et al., 2021) for Generated image with the *keywords* that represent the user's preference. The keywords can be summarized using LLMs based on the user's past interactions. It measures how the generated image aligns with the user's preference.

**Text-Alignment.** It calculates the CLIPscore (Hessel et al., 2021) between the generated image and the user's original prompt, and gauges how the generated image matches the user's intended description.

**Image-Alignment.** It measures the similarity between either the generated image and the ground-truth image. Image-alignment quantifies how closely the current created image aligns with the user's truly saved image.

In the online evaluation, we create an online system to gather real user feedback. Positive feedback ratio is used to evaluate our personalized text-to-image generation model.

## 5.2 IMPLEMENTATION DETAILS

We use ChatGPT (OpenAI) as our rewriter and Stable Diffusion (SD) v1-5 (Rombach et al.) as our text-to-image generation model. SD is sampled by using PNDM scheduler in 50 steps and setting the classifier-free guidance scale to 7.0.

In retrieval, we choose the relevant text prompt number as $k = 3$. For in-context rewriting, we set $L = 5$ and randomly select one demonstration example for each rewriting task, unless otherwise specified.

When calculating DHP, we randomly select 20 history prompts from $\mathcal{Q}_t$ at most for users with prompts larger than 20 image-prompt histories.

Unless other specified, all the experiments using EBR to retrieval hisotrical prompts, and one-short in-context learning to rewrite the shortened prompt.

## 5.3 QUALITATIVE ANALYSIS

To illustrate the effectiveness of our retrieval methods, we provide visualizations of the retrieved relevant image-prompt pairs in the left section of Figure 6. These examples are drawn from the experimental results of three different test prompts from three users. The retrieved histories exhibit a high degree of similarity with their corresponding queries in terms of objects, attributes, as well as the overall style or mood of the image.

This effectively demonstrates the proficiency of our retriever in sourcing relevant user histories, thereby providing a robust reference for our rewriter to carry out personalized prompt rewriting.

In the right section of Figure 6, we display the generation outcomes of both "Shortened Prompt" and "Personalized Prompt". The "Shortened Prompt" column exhibits the results produced from the shortened prompts, while the "Personalized Prompt" column features the rewritten prompts along with their corresponding generated images. It's evident from these displays that our generated im-

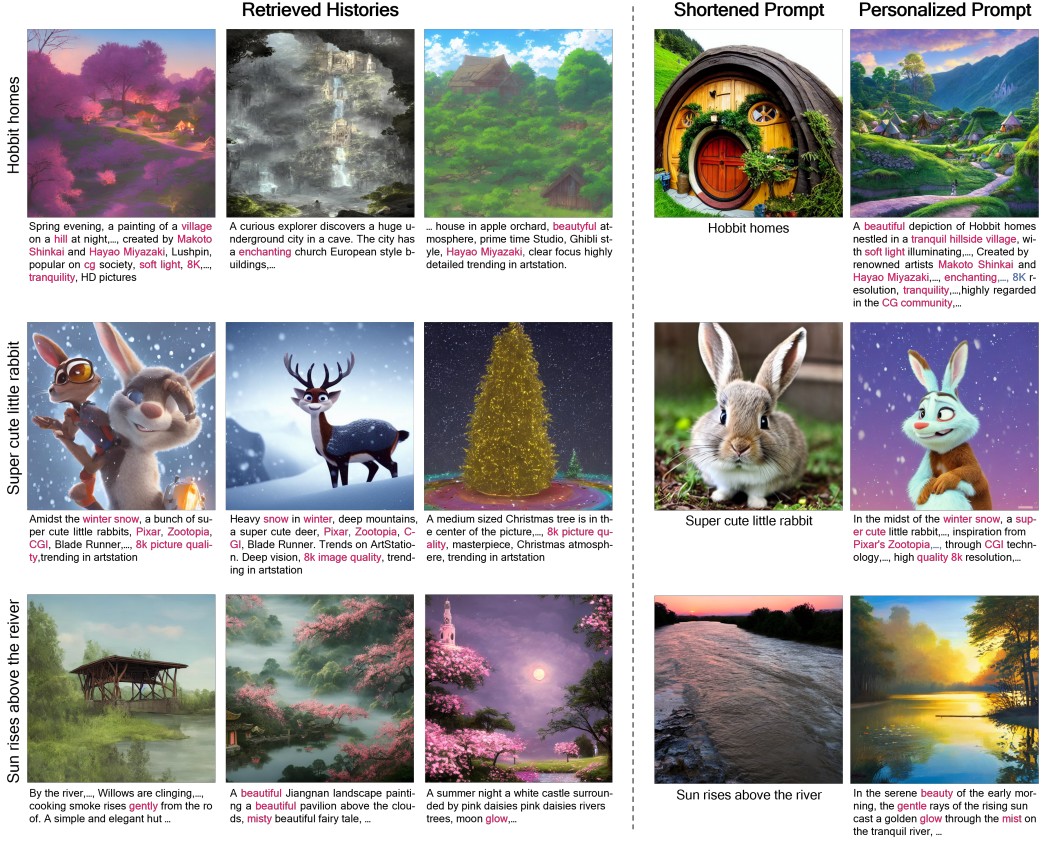

Figure 6: Qualitative analysis of personalized retrieval and rewriting.

ages are more inclined towards user preferences based on their histories, a testament to the expressive power of our rewritten prompt. For instance, when the query "Hobbit homes" is used (as seen in the first row), we observe the user's preferred style across three images, all capturing the mood of mountainous scenery and depicting the Hobbit homes within a consistent landscape.

## 5.4 QUANTITATIVE COMPARISON

To further examine the effectiveness of our personalized prompting methods, we further conduct offline and online quantitative evaluation, comparing different settings to the baseline methods without enhanced prompt rewritten on the test samples we created.

| Method | Retriever | ROUGE-L↑ | DHP↓ | PMS↑ | Text-Align↑ | Image-Align↑ |
|---|---|---|---|---|---|---|
| Shortened Prompt | - | 0.3268 | 0.6618 | 0.5567 | 0.5856 | 0.6272 |
| Personalized Prompt | BM25 | 0.3942 | 0.5942 | 0.6125 | 0.6188 | 0.6581 |
| | EBR | 0.4137 | 0.5943 | 0.6083 | 0.6232 | 0.6485 |
| Personalized + ICL | BM25 | 0.4417 | 0.5418 | **0.6253** | 0.6219 | 0.6456 |
| | EBR | **0.4686** | **0.5332** | 0.6179 | **0.6411** | **0.6796** |

Table 2: Comparison results of different variants of our method with the baseline. Evidently, our method using EBR retriever (top-3 retrieval) and 1-shot ICL can achieve most best results.

**Offline Test** Table 2 showcases the numerical results comparing various retrieval and rewriting configurations with shortened prompts. A comparison between BM25 and EBR reveals that dense retrieval generally outperforms sparse retrieval, although the difference is relatively small, indicating that both methods can yield satisfactory results.

Further, when contrasting context-independent learning with context-dependent learning, it's evident that ICL produces superior outcomes. By integrating ICL with EBR, we achieve absolute improvements of 14.2%, 12.9%, 6.9%, and 5.6% in terms of ROUGE-L, DHP, PMS, and Text-Align metrics respectively. This underscores the exceptional performance of personalized prompting.

To check how sensitive our prompt rewriting methods with respect to the length of the prompts before rewriting. we assess our rewriting method using two additional types of prompts with shorter lengths, namely "Noun Phrase" and "Noun", as detailed in Table 3.

| Prompt Type | Method | ROUGE-L↑ | DHP↓ | PMS↑ | Text-Align↑ | Image-Align↑ |
|---|---|---|---|---|---|---|
| Noun | Original | 0.1770 | 0.7183 | 0.5537 | 0.5330 | 0.6087 |
| | Personalized | 0.2804 | 0.5639 | 0.6142 | 0.5937 | 0.6478 |
| Noun Phrase | Original | 0.2459 | 0.6722 | 0.5554 | 0.5623 | 0.6146 |
| | Personalized | 0.3387 | 0.5557 | 0.6168 | 0.6087 | 0.6534 |
| Short Sentence | Original | 0.3268 | 0.6618 | 0.5567 | 0.5856 | 0.6272 |
| | Personalized | 0.4686 | 0.5332 | 0.6179 | 0.6411 | 0.6796 |

Table 3: Performance with respect to different prompt lengths, i.e. *Noun*, *Noun Phrase* and *Short Sentence*. Our "Personalized Prompt" method equipped with top-3 dense retriever, and 1-shot ICL consistently enhance the results, even with only nouns or noun phrases.

These two prompts are derived using spaCy (Honnibal et al., 2020) from our dataset, adhering to the principle of minimizing word count while maintaining the main entities. The results displayed in Table 3 show that across all three shortened scale, "Personalized Prompt" outperforms the baseline across all metrics, demonstrating the effectiveness of our methods in recovering user preferences.

**Online Test** To further validate our approach, we established an open-domain online evaluation system using SD1.5. Users from the PIP dataset collection platform were invited to participate. Each user input a prompt and compared two images - one from a "Personalized Prompt" and another from an "Original Prompt", choosing their preferred one. We used data samples from the 9th comparison onwards to ensure sufficient history for the "Personalized Prompt". Together, we gathered 311 data samples from 95 users. Our approach was favored in 61.7% of the results, indicating a stronger alignment with user preferences and affirming the effectiveness of our method. We anticipate even better results in real-world scenarios with more historical interactions and a better text to image generation method. Further details on the evaluation method and the platform we used can be found in Appendix A3.

## 6 CONCLUSION AND FUTURE WORK

In conclusion, this paper has presented a novel perspective on large pretrained models, viewing them as search engines, which has enabled us address a significant challenge of creating personalized text-to-image generation.

Our approach has been to enhance user prompts by leveraging historical user interactions with the system. We have proposed a novel method that involves rewriting user prompts based on a new large-scale text-to-image dataset with over 300,000 prompts from 3,115 users. This approach has been shown to enhance the expressiveness and alignment of user prompts with their intended visual outputs. Our experimental results have demonstrated the superiority of our methods over baseline approaches. This was evidenced in our new offline evaluation method and online tests, validating the effectiveness of our approach.

Although the outcomes are encouraging, considerable research still lies ahead in this domain. In the realm of personalized text-to-image generation, the incorporation of more personal details like a user's age and gender could bolster performance. The methodologies used could be extended to other large pretrained models, including language models. Additionally, techniques for enhancing search engines, such as data source purification and ranking optimization, could be assimilated into these models.

We are confident that our contributions represent an advancement towards a more personalized and user-focused artificial intelligence.

## 7 ETHICS STATEMENT

In this section, we discuss three primary ethical considerations regarding both our PIP dataset and the proposed method.

- **Copyright.** When using our text-to-image generation website, all users consent to our privacy policy and data usage agreement. This includes academic utilization and the dissemination of any content hosted on the server, such as anonymized prompts, images, and user iterations.

- **Privacy.** We've ensured that all user data in the PIP dataset, including usernames and other private details, has been anonymized. It is worth noting that while rare, there is a possibility that certain prompts could contain sensitive information. But, this is uncommon because our website has strict rules against writing personal information in the prompts, and has professional detectors to eliminate any violative content as well.

- **Potential Harm.** Although we have built strong detectors to filter out inappropriate material on our website, there remains a potential for occasional instances of NSFW (Not Safe for Work) content to emerge in PIP dataset. Furthermore, our method uses Stable Diffusion v1-5 and ChatGPT as a prior, therefore it inherits any problematic biases and limitations that they may have.

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

# Appendix

## A1 ABLATION STUDY

In this section, we ablate the $k$ relevant historical prompts used to rewrite personalized prompts and the number of demonstration examples used for in-context learning. We evaluate each experiment on the same setup as in Section 5.1 and 5.2.

| Method | Retrieval Top-$k$ | ROUGE-L↑ | DHP↓ | PMS↑ | Text-Align↑ | Image-Align↑ |
|---|---|---|---|---|---|---|
| Personalized + ICL | 1 | 0.4539 | 0.5500 | 0.6057 | 0.6360 | 0.6751 |
| | 3 | **0.4686** | **0.5332** | 0.6179 | **0.6411** | **0.6796** |
| | 5 | 0.4474 | 0.5434 | 0.6204 | 0.6405 | 0.6748 |
| | 7 | 0.4592 | 0.5368 | **0.6265** | 0.6297 | 0.6651 |

Table A1: Ablation for Retrieval Top-k. We empirically conduct experiments with respect to $k \in \{1, 3, 5, 7\}$, keeping the configuration of DENSE retriever and 1-shot ICL.

**Retrieval Top-k Ablation.** As shown in Table A1, when using 3 most relevant historical prompts for rewriting, we can obtain most best results among all evaluation metrics. We analyze that too more historical prompts could provide redundant information and also too long prompt input to ChatGPT worsens the rewriting performance. Therefore, we choose 3 as the number of retrieval results, a balanced manner between performance and efficiency.

| Method (Retrieval) | ICL Shot | ROUGE-L↑ | DHP↓ | PMS↑ | Text-Align↑ | Image-Align↑ |
|---|---|---|---|---|---|---|
| Personalized (BM25) | 1 | 0.4417 | 0.5418 | 0.6253 | 0.6219 | 0.6456 |
| | 3 | 0.4381 | 0.5379 | 0.6289 | 0.6183 | 0.6580 |
| | 5 | 0.4226 | **0.5324** | 0.6236 | 0.6188 | 0.6571 |
| Personalized (EBR) | 1 | **0.4686** | 0.5332 | 0.6179 | **0.6411** | **0.6796** |
| | 3 | 0.4354 | 0.5392 | **0.6274** | 0.6349 | 0.6708 |
| | 5 | 0.4439 | 0.5371 | 0.6242 | 0.6387 | 0.6724 |

Table A2: Ablation for Number of ICL Demonstrations. We experiment with consistent top-3 retrieval both on BM25 and EBR. When we set ICL shot as 1 and use EBR retriever, we observe more superior results appearing.

**Number of ICL Demonstrations Ablation.** Table A2 shows that 1-shot setting, i.e., given 1 demonstration example for in-context learning, can achieve the best results in general among all the four evaluation metrics regarding both BM25 and EBR. This demonstrates that our prompt rewriting template is efficient and effective enough for extracting the personalized preference from numerous historical data of each user.

---

**ChatGPT Input Example**

---

Prompt in text-to-image generation describes the detailed attributes of the object user plans to draw. User's preference in text-to-image generation is shown in history prompts.

Given 3 history prompts, your task is to rewrite the current prompt so that it matches the user's preference. The rewritten prompt should retain primary objects in the original prompt and conform to the user's preference. Please avoid being too diffused and restrict your output within 70 words.

The history prompts are:

   1. *Fusion of nature and technology, high, rise buildings, science and technology cities, sky gardens, ultra, fine, cosmic light, Unreal Engine 5, intricate design, wide angle, imposing, futuristic concept, Thomas Cole, trending in artstation.*

   2. *Tech City, Grand Scenes, Soldiers Under the City, Wide Angle, Complex Backgrounds, Wireless Connectivity, Ambient Light, Cosmic Light, Unreal Engine 5, Cinematic Rendering, trending in artstation*

   3. *Castle, streets, demons, soldiers under the city, wide angle, complex background, wireless connectivity, ambient light, cosmic light, Unreal Engine 5, cinematic rendering, reference to Lord of the Rings style,John Harris,trending in artstation.*

The current prompt is: *Skyscrapers*

The rewritten prompt (one sentence less than 70 words) is:

---

Table A3: An input example for context independent rewriting.

## A2 Details for Personalized Prompt Rewriting

Here, we show details of the two strategies of personalized prompt rewriting we designed in Section 3.2. Table A3 shows an example input for context independent prompt rewriting. The input consists of an instruction we created, the user prompt, and the retrieved relevant prompts $\mathcal{R}_t$. We use *Italic* style font to show concrete contents of $x_t$ and $\mathcal{R}_t$ on the Table A3 and Table A4.

For in-context rewriting, we insert a few demonstration examples between the instruction and the current query. Each demonstration example consists of a user prompt, three relevant history prompts, and a target prompt. Table A4 shows an example input for context dependent prompt rewriting when using one demonstration example.

---

**ChatGPT Input Example**

---

Prompt in text-to-image generation describes the detailed attributes of the object user plans to draw. User's preference in text-to-image generation is shown in history prompts.

Given 3 history prompts, your task is to rewrite the current prompt so that it matches the user's preference. The rewritten prompt should retain primary objects in the original prompt and conform to the user's preference. Please avoid being too diffused and restrict your output within 70 words.

Example:

The history prompts are:

   1. *Best Quality, Masterpiece photo of a girl, Single Person, Perfect Eyes, Acquire Face, Acquire Skin, Black Hair, Long Hair, Necklace, Looking at Camera, City, Street, Night, Hotel, Skyscraper, Skyline, Top Shoulder, Bottom Split Short Skirt, Black Socks, A Pair of Red High Heels, 4k Ultra Clear, Highly detailed, pro Professional digital painting, Unreal Engine 5, Photorealism, HD quality, 8k resolution, cinema 4d, 3D, cinematic, professional photography, art by art and greg rutkowski and alphonse mucha and finish and WLOP,Photography*

   2. *Photography exquisitely portrays realistic characters in the style of Pino daeni, a female college student fashion masterpiece, a classic masterpiece, a close,up of a beautiful girl, black hair, side bun, red dress, happy, long hair fluttering, full moon, dark night, glowing red fireflies, amidst plum blossoms, forest in the distance, colorful and colorful oil painting techniques, the best composition,Photography, realistic.*

   3. *A beautiful photo of a European girl, exquisite and complex skirt, colorful clothing, mature and beautiful, with clear and moving eyes, sitting outdoors, (masterpiece), best quality, midjournal portrait, masterpiece, close,up, by Paul Hedley.*

The current prompt is: *A beautiful girl*

The rewritten prompt is: *A delicate and beautiful girl with a pure temperament, red lips, black hair, straight hair, wearing a black off shoulder top, red split mini skirt, high heels, modern urban clothing. Standing on the evening street, with trees on both sides of the street and leaves falling all over the street, the whole body is photographed, realistic, 8k, The best quality, masterpiece, highlights, beautiful.*

The history prompts are:

   1. *Soft colors beautiful lines, smart lines, flowers surround the edge of the picture, Goddess of Light, Daphne, artist Musa, trending in artstation.*

   2. *Painting by artist Muxia, Flowers and Women, vivid colors, trending in artstation.*

   3. *One person wears a luxurious long dress, a large round window and the scene is magnificent. The European court is extremely luxurious and the cinematic backlight is an epic scene. Sylvain Sarrailh, trending in artstation.*

The current prompt is: *Girl by the window*

The rewritten prompt (one sentence less than 70 words) is:

---

Table A4: An input example for context dependent rewriting.

## A3 Details for Online Evaluation

In this section, we elaborate on the process of online evaluation. Figure A1 displays screenshots of the online evaluation platform we have developed.

For every prompt a user inputs, two generated images are presented. One image is produced using the "Original Prompt" approach, while the other is created using the "Personalized Prompt" method. The user's task is to choose the image they prefer. Our platform then automatically records which method the user favored in each instance.

To ensure that there is enough user interaction history, each user is asked to generate at least 9 images, whwere the first 8 form the user's history. Subsequently, the user's preferred images are added to their history in the following rounds.

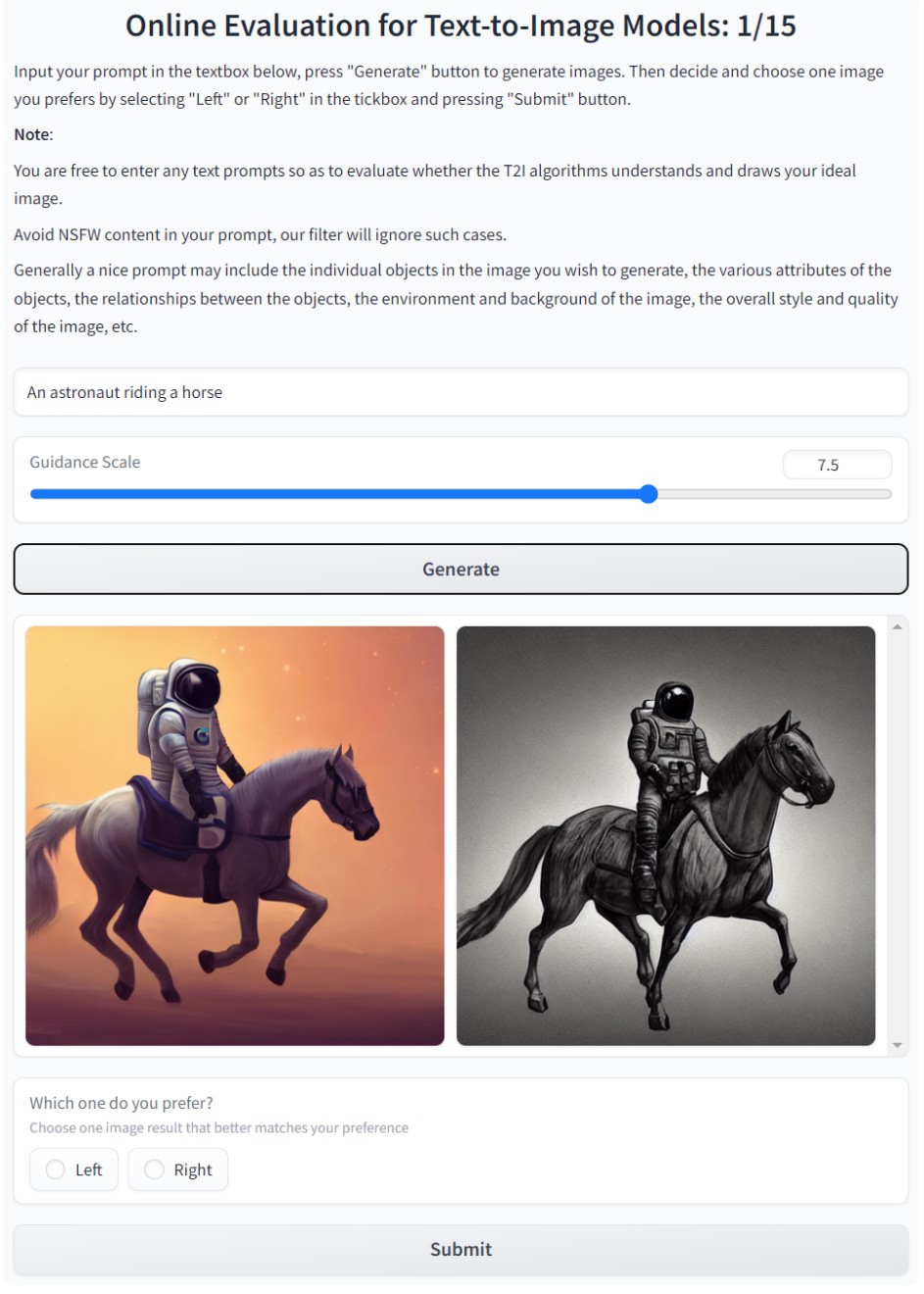

Figure A1: Screenshot of Online Evaluation Platform

