# OpenReview forum: "Tailored Visions: Enhancing Text-to-Image Generation with Personalized Prompt Rewriting"
_ICLR.cc/2024/Conference — ICLR 2024 Conference Withdrawn Submission_

### Official Review · Reviewer_YATF · 2023-10-26

**Soundness:** 1 poor
**Presentation:** 2 fair
**Contribution:** 2 fair
**Rating:** 3
**Confidence:** 4

**Summary:**

This paper proposes a personalized prompt rewriting framework based on a retrieval system and a LLM-based prompt rewriting module. Accordingly, this paper also constructs a prompt dataset with specific users. Some qualitative and quantitative results show the approach's effectiveness.

**Strengths:**

1. This paper proposes to construct a prompt dataset with a specific user ID, which is very important and valuable.
2. Some qualitative and quantitative results show the approach's effectiveness.

**Weaknesses:**

1. This paper means to leverage "historical user interactions with the system to enhance user prompts" and accordingly build a 300k prompts dataset from 3115 users. My main concern is how to determine the user ID when using the system. When I type a prompt, how can you identify my "historical interactions" with the system? In addition, what if I never interact with the system so that you don't have any "historical" information of mine? Further, is historical information about the same user indeed what he or she needs?

2. This paper is just a simple combination of two modules: a retrieval system by CLIP or BM25 and a LLM-based rewriting system by ChatGPT. I am not very clear about the contribution and innovation of your method.

3. There are many pinoneer works that have addressed prompt rewriting issues for text-to-image generation, such as Prompist[1], SUR-adapter[2] and [3-6]. Even though they are rule-based or learning-based methods that don't introduce the so-called personalized prompts concept, you do not compare your approach with any of these works in the experiment section. So how can you prove the effectiveness of your "personalized" approach?

[1] Hao, Yaru, et al., "Optimizing prompts for text-to-image generation." arXiv preprint arXiv:2212.09611 (2022).

[2] Zhong, Shanshan, et al., "Sur-adapter: Enhancing text-to-image pre-trained diffusion models with large language models." arXiv preprint arXiv:2305.05189 (2023).

[3] Oppenlaender, Jonas. "A Taxonomy of Prompt Modifiers for Text-To-Image Generation. arXiv." arXiv preprint arXiv:2204.13988 (2022).

[4] Pavlichenko, Nikita, and Dmitry Ustalov. "Best prompts for text-to-image models and how to find them." Proceedings of the 46th International ACM SIGIR Conference on Research and Development in Information Retrieval. 2023.

[5] Liu, Vivian, and Lydia B. Chilton. "Design guidelines for prompt engineering text-to-image generative models." Proceedings of the 2022 CHI Conference on Human Factors in Computing Systems. 2022.

[6] Wen, Yuxin, et al. "Hard prompts made easy: Gradient-based discrete optimization for prompt tuning and discovery." arXiv preprint arXiv:2302.03668 (2023).

**Questions:**

Please refer to the weakness part.

---

### Official Review · Reviewer_7uv5 · 2023-10-31

**Soundness:** 3 good
**Presentation:** 3 good
**Contribution:** 2 fair
**Rating:** 3
**Confidence:** 4

**Summary:**

The paper focuses on generating personalized prompts for users to create corresponding personalized images using text-to-image generation models. To accomplish this, the authors collect a large-scale dataset and also employ existing methods, including CLIP, stable diffusion, and ChatGPT, to rephrase the prompts and subsequently generate personalized images based on the rephrased prompts.

**Strengths:**

1. The writing is clear and easy to follow.

2. The authors have assembled a personalized prompt dataset, which may be made public upon paper acceptance.

3. The exploration of utilizing large generative models to enhance performance is intriguing.

**Weaknesses:**

1. The paper's objective is to generate personalized prompts for different users, which are then used to produce personalized images using pre-trained text-to-image generation models. However, this personalized process is based on a dataset created from 3,115 users, which seems contradictory to the goal of producing personalized prompts and images for individual users. This aspect needs clarification in the paper.

2. I also have concerns about the concept of retrieving historical prompts. In a real-world application, it might not always be easy for users to have a sufficient quantity of fine-grained and personalized prompts in their history. How can the proposed method be effectively employed in the search engine scenario mentioned by the authors? In such cases, wouldn't it be more practical to directly update the prompts provided to ChatGPT to generate user-desired prompts?

3. Essentially, the proposed method relies on existing techniques, such as the text encoder from CLIP or Stable Diffusion in retrieval and ChatGPT in rewriting. One of the main contributions appears to be the dataset. However, as mentioned in the first point, the collected dataset may not truly represent "personalized information" for various users, which may diminish the overall value of this contribution.
.

**Questions:**

Please see above weaknesses.

---

### Official Review · Reviewer_hJxe · 2023-11-01

**Soundness:** 2 fair
**Presentation:** 1 poor
**Contribution:** 2 fair
**Rating:** 3
**Confidence:** 4

**Summary:**

This paper introduces a new dataset called Personalized Image-Prompt (PIP), which consists of multiple user-specific image-prompt pairs, and presents a method for rewriting personalized text prompts based on user histories.

**Strengths:**

- This paper proposes an approach to rewriting prompts tailored to users based on their history, which is interesting.
- This paper suggests future research direction for personalized and user-centric artificial intelligence.

**Weaknesses:**

- The presentation of the paper is poor, making it hard to read. I couldn't clearly discern the main goal of the paper from the introduction section.
I couldn't understand how this paper views the large foundation model from the perspective of a search engine. As far as I understand, the authors simply ask ChatGPT to generate prompts that reflect user preferences. This does not provide clear evidence of viewing the large foundation model as a search engine.
- The proposed evaluation methods were not justified. (e.g, see questions below )
- This paper lacks comparisons with other approaches. The paper could benefit from a more comprehensive comparison with other existing approaches in the field.
- The problems and dataset presented in the paper are interesting, but researches involving retrieval and rewriting are quite common, so the idea may not be particularly novel.

**Questions:**

- In the evaluation method, such as the Text-Alignment section, the authors calculate the CLIP score between the generated image and the user's original prompt. Given that the main goal of this paper is to rewrite prompts that better encapsulate user preferences based on their history, what is the specific reason for including this evaluation method?

**Details Of Ethics Concerns:**

Authors already mentioned these ethical issues in the paper.